# The Expression Patterns of Immune Checkpoint Molecules in Colorectal Cancer: An Analysis Based on Microsatellite Status

**DOI:** 10.3390/biomedicines12040752

**Published:** 2024-03-28

**Authors:** Sanghyun An, Wanlu Li, Hyejin Do, Hye Youn Kwon, Bora Kim, Kwangmin Kim, Youngwan Kim, Mee-Yon Cho

**Affiliations:** 1Department of Surgery, Yonsei University Wonju College of Medicine, Wonju 26426, Republic of Korea; uldura@yonsei.ac.kr (S.A.); kwonhy@yonsei.ac.kr (H.Y.K.); lukelike@yonsei.ac.kr (K.K.); 2Department of Pathology, Jiaxing University College of Medicine, Jiaxing 314001, China; 3Department of Anesthesiology, Yonsei University Wonju College of Medicine, Wonju 26426, Republic of Korea; dohyejin@yonsei.ac.kr; 4Wonju Surgical Research Collaboration, Wonju 26465, Republic of Korea; 5Department of Internal Medicine, Yonsei University Wonju College of Medicine, Wonju 26426, Republic of Korea; md05bkim@yonsei.ac.kr; 6Trauma Center, Wonju Severance Christian Hospital, Wonju 26426, Republic of Korea; 7Department of Pathology, Yonsei University Wonju College of Medicine, Wonju 26426, Republic of Korea

**Keywords:** colorectal neoplasms, microsatellite instability, immune checkpoint proteins, immunotherapy

## Abstract

Recently, immunotherapy has arisen as a novel treatment approach for patients with colorectal cancer (CRC), but the effectiveness of immunotherapy varies in these patients. We hypothesized that immune checkpoint molecules (ICMs), which are the targets of immunotherapy, are often exhibited concomitantly. Our objective was to investigate the patterns of ICM expression in patients with CRC and the differences in ICM expression based on microsatellite instability status. The immunohistochemical expression of programmed cell death protein 1 (PD-1), programmed cell death ligand 1 (PD-L1), T cell immunoglobulin and mucin-domain containing-3 (TIM-3), and lymphocyte-activation gene 3 (LAG-3) in the tumor center and periphery was assessed in patients with non-metastatic colorectal cancer. We enrolled 83 patients with CRC: a total of 40 microsatellite-stable (MSS) and 43 microsatellite-instability-high (MSI-H) cancer patients. PD-L1 was more frequently expressed in the tumor center in the MSI-H patients with than that in the MSS patients (18 [41.9%] vs. 3 [7.5%], respectively; *p* < 0.001), and the same trend was observed for TIM-3 expression (30 [69.8%] vs. 19 [47.5%], respectively; *p* = 0.047). The concomitant expression of two or more ICMs was more frequently observed than no expression or the expression of a single molecule in both the MSS and MSI-H groups; a total of 34 (79.7%) patients with MSI-H cancer and 23 (57.5%) with MSS cancer showed ICM expression at the tumor center, whereas 34 (79.7%) patients with MSI-H cancer and 22 (55%) with MSS cancer showed expression at the tumor periphery. Patients with the genetic characteristics of MSI-H cancer showed higher expression levels of ICMs than those in patients with MSS cancer, and predominantly, two or more ICMs were concurrently expressed. Our findings highlight the potential efficacy of the dual-blockade approach in immunotherapy, particularly in patients with MSI-H CRC.

## 1. Introduction

Radical surgical resection stands as the principal therapy for colorectal cancer (CRC), and for advanced CRC, the use of cytotoxic chemotherapy and target agents has played a crucial role in treating the disease. However, currently, CRC has an overall recurrence rate of 25%, and metastatic lesions are found in 20% of patients at the time of diagnosis [1,2]. Overall, the treatment of recurrent or metastatic CRC remains a challenge.

Recently, immune checkpoint inhibitors (ICIs), which inhibit immune checkpoint molecules (ICMs), have emerged as novel therapeutic choices for patients with CRC. The interactions between tumors and tumor-specific T cells via ICMs have been identified as key underlying mechanisms within the tumor microenvironment [3,4,5]. Genetic and epigenetic changes during carcinogenesis induce the expression of multiple tumor antigens; however, tumors can survive the immune response by inducing tolerance [6,7]. Immunotherapy using ICIs can counteract tumor immune escape by inhibiting immune checkpoint pathways and reactivating the body’s capacity to initiate an effective antitumor immune response [8]. The effectiveness of ICIs has been proven in other malignant diseases such as melanoma, lung cancer, and prostate cancer [9,10,11].

Although immunotherapy is effective in treating patients with CRC, it specifically displays partial efficacy in a subset of individuals who have the genetic features of microsatellite instability, which accounts for approximately 10% of patients with CRC [12,13,14]. This may be attributed to enhanced tumor resistance due to a lack of tumor-infiltrating lymphocytes and the presence of inhibitory myeloid cells. Additionally, low expression levels of PD-L1, which is the primary target of immunotherapy, in tumors may restrict the effectiveness of treatment [15,16,17,18,19,20,21]. Assessing the expression patterns of ICMs other than PD-L1, which can be targeted using ICIs, and tailoring treatment based on these expression patterns can potentially enhance the therapeutic outcomes in patients with CRC. Immunotherapy, which focuses on multiple ICMs simultaneously, is already being studied and applied to various cancer types, including CRC [22,23].

Therefore, our objective was to study the expression patterns of ICMs via immunohistochemical staining in patients with CRC and assess whether the expression patterns of the identified markers correlated with the clinicopathological characteristics. By analyzing the expression patterns of ICMs, we intended to obtain theoretical support for the multi-blockade of ICMs in CRC treatment. 

## 2. Materials and Methods

### 2.1. Patient Selection

We enrolled individuals diagnosed with stage I–III CRC, according to the American Joint Committee on Cancer (AJCC) guidelines, who underwent curative surgical resection between January 2006 and December 2017. Patients who received only palliative bypass surgery without radical resection, those with cancers in other organs, and those with a history of CRC were excluded. The Institutional Review Board of Wonju Severance Christian Hospital approved this study (approval number: CR:318334), and every participant gave their written informed consent. The study was conducted in accordance with the ethical principles of the Declaration of Helsinki.

### 2.2. Data Collection: Clinicopathological Data

The patients’ clinical information, including age, sex, medical history, the classification recommended by the American Society of Anesthesiologists (ASA), tumor location, carcinoembryonic antigen (CEA) levels, pathologic stage, tumor differentiation, and lymphatic invasion, was obtained from medical records. Low-grade tumors were defined as poorly differentiated mucinous tumors, whereas high-grade tumors were defined as well- or moderately differentiated tumors. Patient survival data were obtained from the colorectal cancer databases of Wonju Severance Christian Hospital and the Korean National Cancer Center.

### 2.3. Tissue Sample Preparation and Microarray Analysis

Paraffin-embedded, formalin-fixed specimens of 83 cases of CRC were acquired from the records of the Department of Pathology at Wonju Severance Christian Hospital (2006–2017). Tissue microarrays (TMAs) were constructed by reviewing the archived tumor slides. A core with a 0.5 mm diameter was inserted into the paraffin block; for each patient, one sample each was obtained from the tumor center and another from the margin of the tumor invasion. Each core was processed into blocks in a 1 mm spaced grid covering approximately 1 square inch, and 5-micron-thick sections were cut. The number of tissue cores in the TMA was 5 × 4. The tissues were arrayed using a manual arraying instrument (Unitma, Republic of Korea).

### 2.4. Immunohistochemical Analysis

The paraffin-embedded tissue samples were sectioned from each block and warmed at 75 °C for 4 min. The slides underwent deparaffinization with EZPrep (Ventana Medical Systems, Tucson, AZ, USA), followed by antigen retrieval for 60 min using a cell-conditioning solution provided by Ventana Medical Systems. To reduce the endogenous peroxidase activity, the sections were treated with an ultraviolet inhibitor for a duration of 4 min. Following this, the sections were rinsed repeatedly with reaction buffer at ambient temperature. For the purpose of immunohistochemical (IHC) staining, the ultraView Universal DAB Detection Kit provided by Ventana Medical Systems was employed. The sections were incubated with primary antibodies against programmed cell death ligand 1 (PD-L1; 1:200; Cell Signaling Technology, Danvers, MA, USA; catalog no.: 13687), programmed cell death protein 1 (PD-1; 1:200; Cell Signaling Technology; catalog no.: 86163, Danvers, Beverly, USA), T cell immunoglobulin and mucin-domain containing-3 (TIM-3; 1:100; Abcam, Cambridge, UK; catalog no.: ab185703,), and lymphocyte-activation gene 3 (LAG-3; 1:200; LifeSpan BioSciences, Seattle, WA, USA; catalog no.: LS-C340186-100) for 2 h at 37 °C in an autostainer (Benchmark XT; Ventana Medical Systems). Subsequently, the slides were washed using the reaction buffer, and a series of reagents from Ventana Medical Systems, including HRP UNIV MULT, DAB, and DAB H2O2, were applied in sequence to each slide, allowing 8 min for each reagent, with intermittent washing using the reaction buffer. A copper solution was then applied to the slides for 4 min. A bluing reagent was added subsequently, after which the slides were again washed with the reaction buffer. For nuclear staining, the sections were counterstained with hematoxylin.

The protein expression was manually analyzed in selected hotspot fields. PD-L1, PD-1, and TIM-3 showed membrane expression, whereas LAG-3 was mainly localized in the cytoplasm. The PD-L1 expression level was defined as either low (≤40%) or high (>41%) [24]. A high degree of lymphocyte infiltration in the PD-1-expressing tumors was defined as a measure of >1.43 tumor-infiltrating lymphocytes per square millimeter [25]. The TIM-3 expression level was evaluated using the HScore system (HScore = stain intensity × percentage of positive cells), and the cut-off points were defined [26,27,28]. The LAG-3 expression level was defined as low (<5%) or high (<5%) [29]. The IHC staining outcomes were independently evaluated by two pathologists who had no knowledge of the patients’ clinical and pathological data. Discrepancies between the pathologists’ findings were resolved according to mutual agreement. The IHC staining results are shown in Figure 1.

### 2.5. Statistical Analysis

The patients were classified into two groups according to their microsatellite status, and a comparative analysis was performed. Categorical variables are presented as frequencies and percentages and were analyzed using the chi-square test. If the expected frequency of the data was less than five, Fisher’s exact test was used. Continuous variables were described as means and standard deviations and were analyzed using Student’s *t*-test. The statistical analyses were carried out using SPSS software (version 25.0; IBM, Armonk, NY, USA), and statistical significance was established at a *p*-value of less than 0.05.

## 3. Results

### 3.1. Differences in the Baseline Characteristics between Patients with Microsatellite-Stable (MSS) and Microsatellite-Instability-High (MSI-H) Cancer

Our analysis included 83 eligible patients, of whom 40 exhibited genetic characteristics consistent with MSS cancer and 43 exhibited MSI-H cancer. The two groups exhibited well-balanced distributions in terms of their age, sex ratio, ASA classification, and pathologic stage; the numbers of women in the MSS and MSI-H groups were 25 (58.1%) and 21 (52.5%), respectively; the mean ages at diagnosis of the patients in the two groups were 66.3 and 67.4 years, respectively; and the proportions of patients with ASA scores of 3 were 16 (37.2%) and 12 (30%), respectively. In each group, 60 and 65.1% of the patients, respectively, were diagnosed with pathologic stage I–II CRC.

In the MSS group, rectal cancer was the most frequent, affecting 17 patients (42.5%), whereas in the MSI-H group, right-sided colon cancer was the most prevalent, accounting for 29 patients (67.4%). Low-grade differentiation, including poorly differentiated and mucinous adenocarcinomas, was more frequent in the MSI-H group than that in the MSS group (12 [27.9%] vs. 2 [5%], respectively; *p* = 0.007). The median follow-up duration for patients was 48 months (interquartile range [IQR]: 30, 73). During the follow-up period, in the MSS group, 11 cases of recurrence were observed (27.5%), whereas in the MSI-H group, five cases of recurrence were identified (11.6%; *p* = 0.067). Regarding cancer-related deaths, two deaths (5%) occurred in the MSS group and five (11.6%) in the MSI-H group, and no statistically significant difference was observed (*p* = 0.435; Table 1).

### 3.2. Differential Expression of Immune Checkpoint Molecules between the MSS and MSI-H Groups 

Assessment of the ICM expression in the center of the tumor tissue in the MSI-H and MSS groups revealed that the expression levels of PD-L1 (18 [41.9%] vs. 3 [7.5%, respectively; *p* < 0.001) and TIM-3 (30 [69.8%] vs. 19 [47.5%], respectively; *p* = 0.047) were significantly higher in the MSI-H group than those in the MSS group. The expression levels of PD-1 (12 [30%] vs. 16 [37.2%], respectively; *p* = 0.643) and LAG-3 (30 [75%] vs. 33 [76.7%], respectively; *p* = 1) were not significantly different between the two groups. At the margin of the tumor tissue, the ICM expression levels were higher in the MSI-H group than those in the MSS group, but the difference was not statistically significant (Table 2).

### 3.3. Concomitant Expression of Immune Checkpoint Molecules

In numerous cases, the concomitant expression of two or more ICMs was confirmed. At the tumor center, the concurrent expression of two or more ICMs was observed in 57 individuals (68.7%). The concurrent expression of all four ICMs was observed in 9 cases (10.3%), whereas high expression levels of a single marker were observed in 16 cases (19.2%). Conversely, the absence of the expression of all four ICMs was observed in 10 patients (12%). In the MSS group, two or more ICMs were co-expressed in 23 individuals (57.5%), whereas in the MSI-H group, 34 individuals (79.1%) demonstrated the co-expression of two or more ICMs (Table 3).

At the tumor border, the concurrent expression of two or more ICMs was observed in 56 individuals (67.5%), and the concurrent expression of all four ICMs was observed in 8 individuals (9.6%), whereas a single marker was highly expressed in 18 individuals (21.7%). No expression of all four ICMs was observed in nine individuals (10.8%). In the MSS group, 22 individuals (55%) exhibited the co-expression of two or more ICMs, whereas in the MSI-H group, 34 individuals (79.1%) demonstrated the co-expression of two or more ICMs (Table 3). The results indicate that patients with CRC showed concomitant expression of various types of ICMs. Notably, in the MSI-H group, a significantly higher degree of concurrent expression of ICMs was observed than that in the MSS group at the tumor border (55 vs. 79.1%, respectively; *p* = 0.03) and tumor center (57.5 vs. 79.1%, respectively; *p* = 0.06) (Table 4 and Figure 2).

## 4. Discussion

Our analysis revealed that the majority of patients with CRC (73 out of 83; approximately 87.95%) exhibited high ICM expression levels. According to microsatellite status, patients with the genetic characteristics of MSI-H cancer showed higher levels of ICM expression. This was particularly noticeable in the tumor center, where the expression levels of PD-L1 and TIM-3 were significantly higher in the MSI-H group than those in the MSS group. Additionally, the expression of ICMs was predominantly characterized by the concurrent expression of two or more markers, accounting for over 68% of cases, as compared to that for single marker expression. Several overlapping expression patterns of TIM-3 and LAG-3 were observed. Remarkably, within both the tumor center and border, the MSI-H group demonstrated higher concurrent expression rates of ICMs than those observed in the MSS group.

Various types of ICMs are involved in malignant diseases [30,31], including PD-1 and PD-L1, which are representative ICMs that are highly expressed in the tumor microenvironment, and ICIs targeting these molecules are utilized in immunotherapy. Representative PD-1 inhibitors approved by the FDA include nivolumab and pembrolizumab, while PD-L1 inhibitors include atezolizumab, avelumab, and durvalumab [32,33,34,35]. Studies have related the overexpression of these markers to the prognosis of tumors, but the conclusions are still debatable. Li et al. reported that the PD-L1 expression in colorectal tumor cells and the PD-1 expression in tumor-infiltrating cells are associated with a favorable 5-year overall survival (OS) and disease-free survival (DFS) in patients with CRC [36]. Conversely, several studies have shown that the overexpression of PD-L1 is associated with adverse clinicopathological characteristics, including poor differentiation, lymphovascular invasion, and poor OS [37,38]. TIM-3 is an immune checkpoint receptor that is predominantly found on the surface of T cells, particularly CD4+ and CD8+ T cells. According to previous studies, TIM-3 is known to be overexpressed in CRC, and its overexpression is correlated with tumor progression [39,40]. Yu et al. documented that increased TIM-3 expression in CRC was correlated with tumor size, TNM stage, and distant metastasis. Furthermore, the suppression of TIM-3 expression substantially decreased cell proliferation, migration, and invasion, and these findings were consistent with those obtained from the analysis of clinical tissues [39]. In addition, several research works on TIM-3 blockade are being conducted for the treatment of various malignant tumors, including colon cancer [41,42]. In this study, we observed higher expression levels of LAG-3 compared to those of other molecules in the tumor center and periphery in both the MSI-H and MSS groups. LAG-3, a surface molecule belonging to the immunoglobulin superfamily, has been detected in tumor-infiltrating lymphocytes (TILs) in various cancer types, including breast, ovarian, and lung cancers. LAG-3 expression is frequently associated with elevated PD-1 expression levels in T cells [43]. LAG-3 is a pivotal checkpoint molecule that potentially exhibits synergistic interactions with PD-1 and PD-L1 during cancer immunotherapy [44,45]. LAG-3, closely linked to CD4, modulates T cell proliferation, activation, and homeostasis. Several studies have indicated that elevated LAG-3 expression in TILs is related to an unfavorable prognosis in various cancers [46,47]. Tavana et al. [48] conducted a study to assess the expression levels of LAG-3 and the prognostic impact of LAG-3 expression on patients with CRC and showed that a high number of LAG-3 + TILs was associated with poor clinicopathological features, especially in patients with left-sided colon cancer. Since LAG-3 was predominantly expressed in patients with CRC in our study and LAG-3 has been regarded as an important prognostic factor, LAG-3 should be regarded as a potential therapeutic target for the treatment of CRC [49] (Table 5).

In this study, we not only observed the expression of single ICMs in different individuals but also observed the concurrent expression of two or more markers in the same individual [30,31]. In our study, the concurrent expression rate of two or more ICMs was over 68%. Additionally, our preliminary study, in which we analyzed the gene expression in tissue samples obtained from 30 patients with CRC using quantitative polymerase chain reaction (qPCR), confirmed the expression of two or more immune-related genes in 29 patients (96.7%) [50]. The concurrent expression of ICMs has been reported not only in CRC but also in other malignant tumors. Datar et al. assessed the expression levels and clinical significance of PD-1, LAG-3, and TIM-3 proteins in non-small cell lung cancer (NSCLC) [51]. They observed the co-expression of ICMs, and this co-expression was linked to T cell activation markers and apoptotic signal receptors. However, the proportion of samples exhibiting dual expression was lower than that of samples displaying a single expression pattern. 

Currently, immunotherapy targeting ICMs is used for the treatment of metastatic and recurrent CRC. However, immunotherapy is only administered to patients with the genetic characteristics of deficient mismatch repair (dMMR)/MSI-H cancer. Furthermore, studies have reported the effectiveness of neoadjuvant immunotherapy in patients with advanced-stage MSI-H CRC [52,53]. The National Comprehensive Cancer Network (NCCN) guidelines recommend neoadjuvant immunotherapy for patients with MSI-H CRC. However, patients with CRC with the genetic characteristics of MSI-H cancer constitute only approximately 5–10% of the total population, limiting the applicability and effectiveness of this treatment. In a phase II clinical trial conducted by Le et al., the partial objective response rate, following the administration of pembrolizumab, for patients with dMMR CRC was 40% and 0% for those with proficient mismatch repair (pMMR) CRC [12]. In a phase II trial conducted by Overman et al., which was the largest immunotherapy trial for CRC, a partial response to nivolumab (an anti-PD-1 antibody) administered either alone or in combination with ipilimumab was observed in 31% of patients with MSI and in 10% of patients with MSS [13]. Additionally, in a phase III study conducted to compare the effects of pembrolizumab and chemotherapy in patients with MSI-H CRC (KEYNOTE-177), the pembrolizumab group demonstrated a greater progression-free survival (PFS) advantage compared to the chemotherapy group; however, the OS did not differ significantly between the two groups [54]. Our results support the theory that ICMs operate via complex mechanisms involving multiple molecules rather than via a single mechanism. 

As noted in our study, in numerous instances, two or more ICMs were expressed concurrently. Therefore, theoretically, blocking two or more ICMs simultaneously would be effective. Numerous studies have focused on assessing the effectiveness of the simultaneous inhibition of two ICMs in CRC. The CheckMate 142 study, a phase 2 single-arm study investigating the efficacy of nivolumab and ipilimumab combination therapy in patients with MSI-H mCRC, showed convincing results in terms of a durable objective response rate (65%), encouraging survival rates, and manageable safety [23,55,56]. Another phase 2 study, the GERCOR NIPICOL study, involving the administration of nivolumab plus ipilimumab in patients with MSI-H mCRC, reported a 12-week disease control rate of 86% and 12-month PFS and OS rates of 72.9 and 84%, respectively [57]. The NICHE study, in which the effect of a dual blockade with nivolumab and ipilimumab in neoadjuvant immunotherapy was investigated, reported that 19 out of 20 patients with dMMR CRC showed a major pathologic response (≤10% residual viable tumor), whereas 3 out of 15 patients with pMMR CRC demonstrated a major pathologic response [52]. Further investigations involving a larger patient cohort are necessary to confirm the efficacy of dual blockade in CRC treatment. Research on the effectiveness of dual-blockade therapy for other malignant diseases has been conducted. Hellmann et al. observed that first-line treatment with nivolumab (anti-PD 1 antibodies) combined with ipilimumab (anti-CTLA4 antibodies) resulted in a longer overall survival than that observed with chemotherapy in individuals with NSCLC, independent of the PD-L1 expression levels [58]. Paz-Ares et al. showed that combining nivolumab and ipilimumab with two cycles of chemotherapy significantly enhanced the overall survival compared to that observed with chemotherapy alone and led to a favorable risk–benefit profile [11]. Additionally, research by Hussein et al. found that targeting both LAG-3 and PD-1 simultaneously in the treatment of advanced melanoma can significantly increase survival rates [59].

The four ICMs investigated in this study showed a higher expression in the MSI-H group than that in the MSS group. Particularly, the PD-L1 expression at the center of the tumor was significantly higher in the MSI-H group than that in the MSS group. This finding aligns with the findings of numerous prior studies. Valentini et al. reported that the proportion of PD-L1-expressing tumor cells and infiltrating immune cells was higher in patients with MSI-H CRC than that in patients with MSS CRC [17]. Ho et al. demonstrated that PD-L1 expression in stromal tumor-infiltrating immune cells and intraepithelial tumor-infiltrating immune cells was more frequent in the MSI-H group than that in the MSS group [18]. Therefore, when selecting an immunotherapy target for colorectal cancer, targeting the MSI-H group rather than the MSS group may show a better therapeutic effect. Additionally, our study revealed that in the MSI-H patient cohort, cases with the concomitant expression of two or more types of ICMs were more prevalent than those showing the expression of a single ICM. To the best of our knowledge, this issue has not been addressed in previous studies.

This study had several limitations. First, the limited sample size and absence of a validated independent cohort could have biased the findings. Second, concerning multiple comparisons, the risk of Type I errors cannot be dismissed. Additionally, IHC staining was conducted on a tissue array, which could introduce a selection bias. Despite these limitations, this study is meaningful because the results provide information on the expression patterns of ICMs. 

In conclusion, patients with the genetic characteristics of MSI-H cancer showed a higher tendency toward ICM expression, and predominantly, two or more ICMs were concurrently expressed. These findings highlight the potential efficacy of the multiple-blockade approach in immunotherapy, particularly in patients with MSI-H CRC. Further investigations with a larger number of participants are necessary, and research must be conducted to obtain fundamental evidence that can advance our understanding and the treatment strategies. 

## Figures and Tables

**Figure 1 biomedicines-12-00752-f001:**
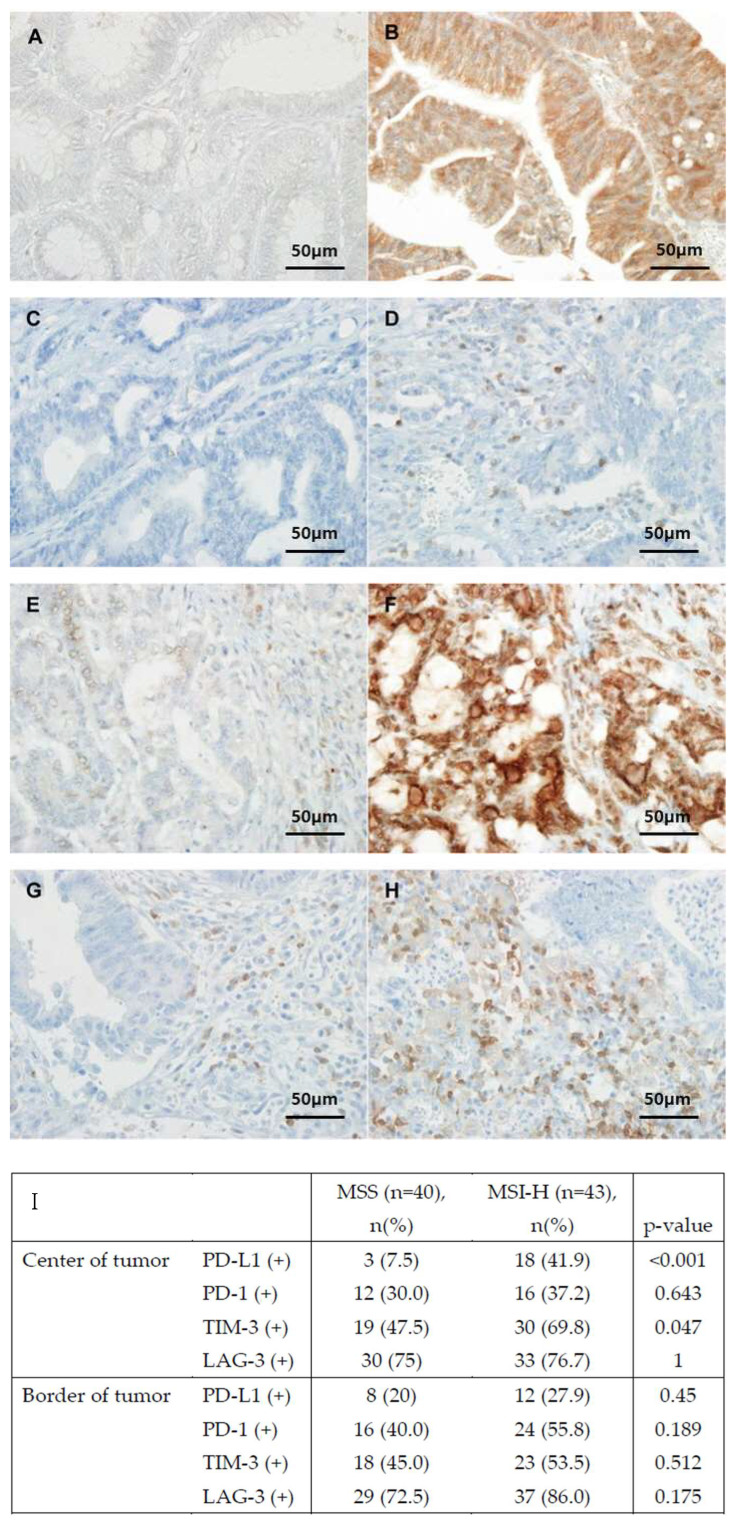
Immunohistochemical analysis of immune checkpoint proteins and the proportion of expression of immune checkpoint molecules (ICMs) in patients with MSS and MSI-H colorectal cancer. (**A**) Negative PD-L1 expression. (**B**) Positive PD-L1 expression in tumor cells. (**C**) Negative PD-1 expression. (**D**) Positive PD-1 expression in tumor cells. (**E**) Negative LAG-3 expression. (**F**) Positive LAG-3 expression in tumor cells. (**G**) Negative TIM-3 expression. (**H**) Positive TIM-3 expression. (**I**) Expression of immune checkpoint molecules (ICMs) in patients with MSS and MSI-H colorectal cancer. PD-L1, programmed cell death ligand 1; PD-1, programmed cell death protein 1; TIM-3, T cell immunoglobulin and mucin domain-containing protein 3; LAG-3, lymphocyte-activation gene 3.

**Figure 2 biomedicines-12-00752-f002:**
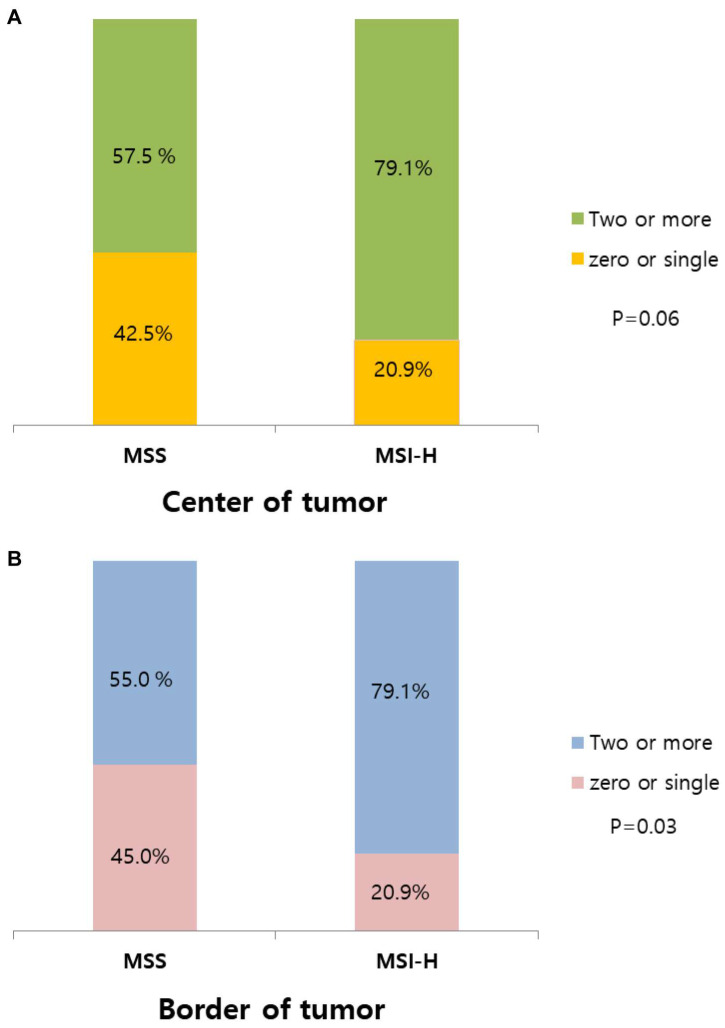
Frequency of concurrent expression of multiple immune checkpoint molecules. (**A**) Center of tumor. (**B**) Border of tumor. In the MSI-H group, a significantly higher degree of concurrent expression of ICMs was observed than that in the MSS group at the tumor border (55 vs. 79.1%, respectively; *p* = 0.03) and tumor center (57.5 vs. 79.1%, respectively; *p* = 0.06). MSS, microsatellite-stable; MSI-H, microsatellite-instability-high.

**Table 1 biomedicines-12-00752-t001:** Comparison of the baseline demographics of patients with colorectal cancer stratified into the MSS or MSI-H groups.

	MSS (%) (n = 40)	MSI-H (%) (n = 43)	*p* Value
Age (year)			
Mean (SD)	67.4 (13)	66.3 (14.5)	0.305
Gender			
Female	21 (52.5)	25 (58.1)	0.662
Male	19 (47.5)	18 (41.9)	
ASA score			
1	5 (12.5)	5 (11.6)	0.784
2	23 (57.5)	22 (51.2)	
3	12 (30.0)	16 (37.2)	
Tumor location			
Proximal	10 (25.0)	29 (67.4)	0.001
Distal	13 (32.5)	7 (16.3)	
Rectum	17 (42.5)	7 (16.3)	
CEA			
<5	26 (65.0)	31 (72.1)	0.636
5≥	14 (35.0)	12 (27.9)	
T stage			
2–3	29 (72.5)	28 (65.1)	0.489
4	11 (27.5)	15 (34.9)	
N stage			
0	25 (62.5)	28 (65.1)	0.823
1–2	15 (37.5)	15 (34.9)	
TNM stage			
I–II	24 (60.0)	28 (65.1)	0.656
III	16 (40.0)	15 (34.9)	
Differentiation			
High-grade	38 (95.0)	31 (72.1)	0.007
Low-grade	2 (5.0)	12 (27.9)	
Lymphatic invasion			
No	22 (55.0)	22 (51.2)	0.827
Yes	18 (45.0)	21 (48.8)	
Lymph node harvest			
Mean (SD)	25.48 (11.79)	33.2 (13.37)	0.320
Recurrence			
No	29 (72.5)	38 (88.4)	0.067
Yes	11 (27.5)	5 (11.6)	
Cancer-related mortality			
No	38 (95.0)	38 (88.4)	0.435
Yes	2 (5.0)	5 (11.6)	

MSS, microsatellite-stable; MSI-H, microsatellite-instability-high; SD, standard deviation; ASA, American Society of Anesthesiologists; CEA, carcinoembryonic antigen.

**Table 2 biomedicines-12-00752-t002:** Expression of immune checkpoint molecules (ICMs) in patients with MSS and MSI-H colorectal cancer.

	MSS (%) (n = 40)	MSI-H (%) (n = 43)	*p* Value
Center of tumor			
PD-L1			
negative	37 (92.5)	25 (58.1)	<0.001
positive	3 (7.5)	18 (41.9)	
PD-1			
negative	28 (70.0)	27 (62.8)	0.643
positive	12 (30.0)	16 (37.2)	
TIM-3			
negative	21 (55.2)	13 (30.2)	0.047
positive	19 (47.5)	30 (69.8)	
LAG-3			
negative	10 (30.0)	10 (23.3)	1.000
positive	30 (75.0)	33 (76.7)	
Border of tumor			
PD-L1			
negative	32 (80.0)	31 (72.1)	0.450
positive	8 (20.0)	12 (27.9)	
PD-1			
negative	24 (60.0)	19 (44.2)	0.189
positive	16 (40.0)	24 (55.8)	
TIM-3			
negative	22 (55.0)	20 (46.5)	0.512
positive	18 (45.0)	23 (53.5)	
LAG-3			
negative	11 (27.5)	6 (14.0)	0.175
positive	29 (72.5)	37 (86.0)	

MSS, microsatellite-stable; MSI-H, microsatellite-instability-high; PD-L1, programmed cell death ligand 1; PD-1, programmed cell death protein 1; TIM-3, T cell immunoglobulin and mucin domain-containing protein 3; LAG-3, lymphocyte-activation gene 3.

**Table 3 biomedicines-12-00752-t003:** Concurrent expression of multiple immune checkpoint molecules.

Center of Tumor	MSS (n = 40)	MSI-H (n = 43)	Total (n = 83)
PD-L1 + PD1 + TIM3 + LAG3	1	8	9
PD-L1 + PD1 + TIM3	0	0	0
PD-L1 + PD1 + LAG3	1	0	1
PD-L1 + TIM3 + LAG3	0	4	4
PD-1 + TIM3 + LAG3	5	3	8
PD-L1 + PD1	0	2	2
PD-L1 + TIM3	0	0	0
PD-L1 + LAG3	1	3	4
PD1 + TIM3	0	1	1
PD1 + LAG3	3	1	4
TIM3 + LAG3	12	12	24
PD-L1	0	1	1
PD1	2	1	3
TIM3	1	2	3
LAG3	7	2	9
All negative	7	3	10
Border of tumor			
PD-L1 + PD1 + TIM3 + LAG3	4	4	8
PD-L1 + PD1 + TIM3	0	0	0
PD-L1 + PD1 + LAG3	4	6	10
PD-L1 + TIM3 + LAG3	0	0	0
PD1 + TIM3 + LAG3	5	6	11
PD-L1 + PD1	0	1	1
PD-L1 + TIM3	0	0	0
PD-L1 + LAG3	0	1	1
PD1 + TIM3	1	1	2
PD1 + LAG3	1	4	5
TIM3 + LAG3	7	11	18
PD-L1	0	0	0
PD1	1	2	3
TIM3	1	1	2
LAG3	8	5	13
All negative	8	1	9

MSS, microsatellite-stable; MSI-H, microsatellite-instability-high; PD-L1, programmed cell death ligand 1; PD-1, programmed cell death protein 1; TIM-3, T cell immunoglobulin and mucin domain-containing protein 3; LAG-3, lymphocyte-activation gene 3.

**Table 4 biomedicines-12-00752-t004:** Expression of multiple immune checkpoint molecules in patients with MSS and MSI-H colorectal cancer.

	Zero or Single	Two or More	*p*-Value
Center of tumor			
MSS	17 (42.5)	23 (57.5)	
MSI-H	9 (20.9)	34 (79.1)	0.06
Border of tumor			
MSS	18 (45.0)	22 (55.0)	
MSI-H	9 (20.9)	34 (79.1)	0.03

MSS, microsatellite-stable; MSI-H, microsatellite-instability-high.

**Table 5 biomedicines-12-00752-t005:** Various types of FDA-approved immune checkpoint inhibitors.

Inhibitor Type	Drug Names
PD-1	Pembrolizumab (Keytruda^®^), Nivolumab (Opdivo^®^), Cemiplimab (Libtayo^®^), Dostarlimab (Jemperli), Retifanlimab (Zynyz)
PD-L1	Atezolizumab (Tecentriq^®^), Avelumab (Bavencio^®^), Durvalumab (Imfinzi™)
LAG-3	Relatlimab (Opdualag™)
CTLA-4	Ipilimumab (Yervoy^®^), Tremelimumab (Imjudo^®^)

PD-1, programmed cell death protein 1; PD-L1, programmed cell death ligand 1; LAG-3, lymphocyte-activation gene 3; CTLA-4, cytotoxic T-lymphocyte associated protein 4.

## Data Availability

Data are contained within the article.

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
