# Peer review of "The Expression Patterns of Immune Checkpoint Molecules in Colorectal Cancer: An Analysis Based on Microsatellite Status"

_biomedicines, 2024, doi:10.3390/biomedicines12040752_

Round 1

Reviewer 1 Report

Comments and Suggestions for Authors

This study analyzed the expression patterns of immune checkpoint molecules in colorectal cancer patients based on tissue microarray. Additionally, the authors suggested differences in the expression patterns of immune checkpoint molecules depending on microsatellite stability. This study presented results that provide insight into the efficacy of immunotherapy according to the stability of microsatellites in colorectal cancer. This manuscript is well written and describes the importance and limitations of the study. I would like to make one comment to improve this manuscript.

1.     I recommend that the authors provide additional immunohistochemistry observations based on the results presented in the table. In addition to the negative/positive expression results of the proteins shown in Figure 1, please provide representative images of patterns of immune checkpoint molecules that are expressed differently depending on microsatellite status.

Author Response

Thank you very much for your valuable comments and for providing us the opportunity to further clarify the findings of our study.

Unfortunately, we don't have representative images of patterns of immune checkpoint molecules depending on microsatellite status.

We aimed to investigate the IHC expression of immune checkpoint molecules using

Thank you very much for your valuable comments and for providing us the opportunity to further clarify the findings of our study.

Unfortunately, we don't have representative images of patterns of immune checkpoint molecules depending on microsatellite status.

We aimed to investigate the IHC expression of immune checkpoint molecules using paraffin blocks. Our analysis revealed that patients with MSI-H demonstrated a higher frequency of overexpression of certain immunologic markers. A comparison of expression patterns in MSI-H vs. MSS, when overexpressed, was not part of the scope of this study. However, we believe that your suggestion is invaluable and will consider this comparison in further study. Thank you once again for your thoughtful review and suggestions.

ng paraffin blocks. Our analysis revealed that patients with MSI-H demonstrated a higher frequency of overexpression of certain immunologic markers. A comparison of expression patterns in MSI-H vs. MSS, when overexpressed, was not part of the scope of this study. However, we believe that your suggestion is invaluable and will consider this comparison in further study. Thank you once again for your thoughtful review and suggestions.

Reviewer 2 Report

Comments and Suggestions for Authors

The study investigated immune checkpoint molecule (ICM) expression patterns in colorectal cancer (CRC) patients, particularly focusing on microsatellite instability-high (MSI-H) tumors. Results revealed higher PD-L1 and TIM-3 expression in the tumor center of MSI-H CRC patients compared to microsatellite stable (MSS) CRC patients. Additionally, concomitant expression of multiple ICMs was more common in MSI-H CRC, suggesting potential efficacy of dual-blockade immunotherapy in this subgroup.

However, the following shortcomings must be addressed:

The ordinate (vertical coordinate) of Fig. 2a represents the percentage of patients expressing programmed cell death ligand 1 (PD-L1) in the tumor center.

The authors should include a list of PD-1 and PD-L1 inhibitors approved by the FDA and under clinical trials for colorectal cancer treatment.

The figures in Fig. 1 should include scales, and statistical analyses need to be added to Fig. 1.

In the legend of Fig. 1, the immunohistochemistry analysis should specify the sample number and include P-value analysis.

A conclusion should be added to summarize the findings and implications of the study.

Comments on the Quality of English Language

 Moderate editing of English language required

Author Response

Comment 1.:  The ordinate (vertical coordinate) of Fig. 2a represents the percentage of patients expressing programmed cell death ligand 1 (PD-L1) in the tumor center.

Reviewed and Corrected: Thank you for your attention to the details presented in our manuscript, particularly regarding Figure 2. We appreciate the opportunity to clarify the interpretation and significance of the data depicted in this figure.

Figure 2 is intended to illustrate not solely the expression of PD-L1, but rather the concurrent expression of various immune checkpoint molecules (ICMs) within the tumor microenvironment. Specifically, our data highlights a significant difference in the frequency of concurrent ICM expression between the microsatellite instability-high (MSI-H) group and the microsatellite stable (MSS) group, both at the tumor border and tumor center. To clarify, in the MSI-H group, we observed a significantly higher frequency of concurrent expression of ICMs compared to the MSS group at the tumor border (55% vs. 79.1%, respectively; P = 0.03) and at the tumor center (57.5% vs. 79.1%, respectively; P = 0.06). This finding underscores the enhanced immune activity within the MSI-H tumors, suggesting a potentially different therapeutic response or prognosis for patients within this group. For the sake of clear understanding among readers, I have added detailed explanations to the legend of Fig 2.

We hope this clarification addresses the concerns raised and accurately conveys the findings of our study.

Comment 2: The authors should include a list of PD-1 and PD-L1 inhibitors approved by the FDA and under clinical trials for colorectal cancer treatment.

à We express our sincere gratitude for your valuable comment which has contributed significantly to the enrichment of our discussion section. Following your suggestion, we have included the following sentence in the discussion to provide clarity on the utilization of immune checkpoint inhibitors in the treatment of malignancies: "Representative PD-1 inhibitors approved by the FDA include nivolumab and pembrolizumab, while PD-L1 inhibitors include atezolizumab, avelumab, and durvalumab." This addition not only reflects the current state of approved therapeutic options but also highlights the relevance of our study within the context of available treatments.

We believe that this revision will enhance the readers' comprehension of the immune checkpoint landscape and the significance of PD-1 and PD-L1 inhibitors in cancer immunotherapy.

Thank you once again for your insightful feedback.

Comment 3: The figures in Fig. 1 should include scales, and statistical analyses need to be added to Fig. 1. In the legend of Fig. 1, the immunohistochemistry analysis should specify the sample number and include P-value analysis.

à We are grateful for your constructive feedback regarding the presentation of immunohistochemical findings in our figures. As per your suggestion, we have now included a scale bar to denote a 400x magnification directly on each image within Figure 1. This addition, along with the updated figure legend, will provide clear guidance on the magnification level and facilitate a more accurate interpretation of the marker expression observed in the tissue sections. Additionally, I specified the sample number of each marker in the legend of Fig.1.

Furthermore, we have carefully considered your recommendation to include statistical analyses in Figure 1. Figure 1 is intended to serve as a visual representation of the histological observations of marker expression. As such, we believe that the incorporation of statistical data within this particular figure might detract from its intended purpose. Instead, we have presented the comparative analysis of the proportion of overexpressed markers between different MSI statuses in Table 2. This allows for a clear and focused presentation of the statistical results, in line with the objectives of our study.

We hope that these revisions meet your approval and enhance the manuscript's quality and readers' understanding.

Comment 4: A conclusion should be added to summarize the findings and implications of the study.

Reviewed and Corrected: We sincerely appreciate your valuable comments and suggestions, which have significantly contributed to enhancing the clarity and impact of our manuscript. Following your advice, we have carefully revised the conclusion section to more accurately reflect the implications of our findings and the potential future directions for research in this area.

à“In conclusion, patients with the genetic characteristics of MSI-H showed a higher tendency for ICM expression, and predominantly, two or more ICMs were concurrently expressed. These findings highlight the potential efficacy of the multiple-blockade approach in immunotherapy, particularly in patients with MSI-H CRC. Further investigations with a larger number of participants are necessary, and research must be conducted to obtain fundamental evidence that can advance our understanding and treatment strategies.”

Thank you once again for your insightful feedback.

Round 2

Reviewer 2 Report

Comments and Suggestions for Authors

While the authors have addressed some concerns in their response, further modifications are required. To enhance the comprehensibility of immune checkpoint research, the authors should provide a clear table delineating all PFA-approved drugs and those undergoing clinical studies. This will offer readers a comprehensive understanding of the current landscape in this field.

As for Figure 1, the authors should employ ImageJ software to process the image, calculate the positive expression rate, and conduct statistical analysis. A high-quality published article should feature detailed and scientifically sound data to persuade readers effectively. Therefore, incorporating these enhancements will elevate the rigor and credibility of the research findings.

Comments on the Quality of English Language

Minor editing of English language required

Author Response

We sincerely thank the reviewers once again for evaluating our manuscript. In response to the second round of revision requests, we have carefully addressed the valuable critiques and suggestions provided, paying close attention to further integrate the reviewers' insights. All changes have been highlighted in green within the document, and a detailed, point-by-point response to the reviewers' comments is included. We deeply appreciate the reviewers' continuous feedback and patience, and we hope that our revisions meet their expectations

Round 3

Reviewer 2 Report

Comments and Suggestions for Authors

The authors well answered the raised questions. This revised version can be published.

Comments on the Quality of English Language

Minor editing of English language required